# Molecular Mechanisms of the Microbiota–Gut–Brain Axis in the Onset and Progression of Stroke

**DOI:** 10.3390/ijms262010071

**Published:** 2025-10-16

**Authors:** Javier Caballero-Villarraso, Sara Pons-Villarta, Jerónimo Cruces-Párraga, Ainoa Navarrete-Pérez, Antonio Camargo, Juan Antonio Moreno, Isaac Túnez, Eduardo Agüera-Morales

**Affiliations:** 1Maimónides Biomedical Research Institute of Córdoba (IMIBIC), 14004 Córdoba, Spain; h92povis@uco.es (S.P.-V.); jeroparraga@gmail.com (J.C.-P.); ep2napea@uco.es (A.N.-P.); antonio.camargo@imibic.org (A.C.); juan.moreno@uco.es (J.A.M.); itunez@uco.es (I.T.); fm2agmoe@uco.es (E.A.-M.); 2Clinical Analyses Service, Reina Sofía University Hospital, 14004 Córdoba, Spain; 3Department of Biochemistry and Molecular Biology, University of Córdoba, 14071 Córdoba, Spain; 4Neurology Department, Reina Sofía University Hospital, 14004 Córdoba, Spain; 5Lipids and Atherosclerosis Unit, Department of Internal Medicine, Reina Sofía University Hospital, 14004 Córdoba, Spain; 6CIBER Fisiopatología de la Obesidad y Nutrición (CIBEROBN), Instituto de Salud Carlos III, 28029 Madrid, Spain; 7Department of Cell Biology, Physiology and Immunology, University of Córdoba, 14071 Córdoba, Spain

**Keywords:** stroke, cerebrovascular disease, blood–brain barrier, microbiota, gut–brain–microbiota axis, stool transplant

## Abstract

The bidirectional relationship between the brain and gut microbiota has led to the concept of the microbiota–gut–brain axis. It refers to a system of bilateral communication that integrates neuronal, immunological, and metabolic signals, whose disruption has been linked to the pathogenesis of digestive, metabolic, and neurological disorders, among others. Intestinal dysbiosis (an imbalance in the gut microbiota) can promote a proinflammatory and prothrombotic state, as well as dyslipidaemia and dysglycemia, that increase atherogenic risk and consequently the risk of stroke. Dysbiosis can also lead to neuroinflammatory and neurodegenerative effects, compromising the integrity of the blood–brain barrier and exacerbating brain injury after stroke. Specific bacterial profiles have been associated with varying levels of stroke risk, emphasising the role of gut microbiota-derived vasoactive metabolites such as Trimethylamine N-Oxide (TMAO), phenylacetylglnutamine (PAGln), and short-chain fatty acids (SCFAs), which may serve as biomarkers for stroke risk and severity. Gut microbiota also influences neurotrophic factors such as brain-derived neurotrophic factor (BDNF) and glial cell-derived neurotrophic factor (GDNF), involved in recovery after stroke. Research has explored the potential to modify the gut microbiota to either prevent stroke (by reducing risk) or improve outcomes (by decreasing severity and sequelae). Current scientific evidence supports the role of gut microbiota as a potential diagnostic and prognostic biomarker, as well as a therapeutic target.

## 1. Introduction

The relationship between health and microbiota has been known for years. The term ‘microbiota’ dates back to the early 20th century, when it was discovered that a large number of microorganisms (bacteria, fungi, archaea, viruses and parasites) coexist with the host at various sites in the body (gut, skin, respiratory tract, skin folds, etc.). The term ‘microbiome’, on the other hand, refers to a much broader aspect, including not only the community of microorganisms but also the ecological habitat or niche, the metabolites secreted by the microbiota, the environmental conditions and the interactions of all these with the host [1,2,3,4].

The microbiota–host relationship is usually one of cooperation or mutualism [5,6,7]. This balance is maintained by what is known in evolutionary biology as ‘imposition’ and refers to mechanisms that evolve to reduce the ‘egocentric’ behaviour of each party within the cooperative or collaborative relationship. In this context, the host develops control mechanisms to suppress harmful behaviours of the microbiota, such as uncontrolled proliferation of pathogens or excessive consumption of host resources, thus ensuring that cooperation is maintained [8,9]. These mechanisms ensuring the balance between microbiota and host are controlled by the constant action of the immune system. For example, an acceleration of intestinal transit that occurs in the event of infection would serve to help eliminate or expel unwanted micro-organisms [10,11,12].

Each microbiota profile or composition can be a unique and subject-specific set of microbial communities, like a specific fingerprint [13,14,15]. Therefore, the scientific community now considers microbiota to be an essential organ for life and to play a relevant role in health and disease states. Moreover, it is now known that the global genome of microbiota is equivalent to 150 times the human genome [16].

Among all the locations of microbial ecosystems in higher organisms, the gut microbiota is undoubtedly the most relevant in diversity and impact compared to other locations in the organism. It develops from before birth and consists of more than 1500 species, distributed in more than 50 different phyla, including bacteria, viruses and some eukaryotic species [17,18,19]. The most predominant phyla are *Bacteroidetes* and *Firmicutes* followed by *Proteobacteria*, *Fusobacteria*, *Tenericutes*, *Actinobacteria* and *Verrucomicrobia*, which together constitute up to 90% of the gut microbial population. In addition, the most frequent bacterial genera are *Bacteroides*, *Clostridium*, *Peptococcus*, *Bifidobacterium*, *Eubacterium*, *Ruminocococcus*, *Faecalibacterium* and *Peptostreptococcus* [20,21,22]. *Bacteroides* alone make up about 30% of the bacteria in the gut, highlighting their relevance to the physiology (or potential pathophysiology) of the host [16,23,24].

However, the microbial composition is not identical throughout the digestive tract. Both the stomach and the small intestine contain a lower relative abundance of bacterial species, whereas the colon is densely populated. Such is their abundance that they contain up to 1012 microbial cells per gram of intestinal substance and constitute up to 60% of the dry mass of faeces. Moreover, while 99% of intestinal bacteria are anaerobic, the cecum contains high densities of aerobic micro-organisms. It should be noted that specific mention has not been made of fungi [25,26], protists [27,28], archaea [29,30] and viruses [31,32,33] as, although they are present in the gut microbiota, their functions are less well understood [16,34,35,36].

The microbiome is a dynamic ecosystem, capable of rapid change depending on intrinsic and extrinsic host factors, which may involve beneficial but also detrimental actions. Its composition has been found to vary with age (reflecting immunosenescence) [37,38,39], gender (mainly mediated by hormonal actions) [40,41], host genetics (in particular the genotype of the immune system) [42,43,44], diet and lifestyle (with marked differences depending on whether they contain a high proportion of fat and protein or are high-fibre diets), antimicrobials, stress and smoking [2,16,45,46].

### Role of the Microbiota in the Health-Disease Coupling

As mentioned above, whether the microbiota and the host are in a situation of balance (or ‘eubiosis’) or imbalance (‘dysbiosis’) can determine whether the individual is healthy or not [47,48]. The microbiota has such important actions as protection against pathogens, immune system development, digestion and metabolism, epithelial cell proliferation and differentiation, modulation of insulin secretion and resistance, and gut–brain communication [16,49,50].

In terms of digestion, these microorganisms metabolise dietary elements and transform them into bioactive compounds that can exert beneficial biological effects. For example, *Firmicutes* and *Bacteroides* are able to metabolise non-digestible carbohydrates (cellulose, resistant starch, pectin, oligosaccharides and ligin) into short-chain fatty acids (SCFAs) such as acetic, propionic and butyric acid [16,51]. In turn, these SCFAs are a primary energy source for the colonic epithelium and reinforce the intestinal mucosal barrier. After absorption, they are used systemically as a substrate for lipogenesis and gluconeogenesis [2]. In addition, it has recently been confirmed that SCFAs have anti-inflammatory and antineoplastic effects, and in subjects with colon cancer, they are lower than in healthy subjects [16,51,52].

The gut microbiota is also involved in the synthesis of B vitamins (biotin, thiamine (B1), cyanocobalamin (B12), riboflavin, nicotine and pantothenic acid) and vitamin K, for which commensal *Bifidobacterium* species are responsible [2,53,54]. With regard to lipid metabolism, it has been reported that the gut microbiota is involved in the synthesis of bile acids, cholesterol and conjugated fatty acids [2].

Microbiota also modulates the activity of the enteric nervous system (ENS), which controls colon motility. When dysbiosis is present, there is a reduction in the excitability of enteric neurons due to insufficient serotonin (5-HT) synthesis and the concentration of SCFAs decreases, which decreases motility and intestinal transit [55,56]. In addition, in dysbiosis there is a greater number of bacteria that produce methane, a neuromuscular transmitter that affects the muscle function of the colon. These mechanisms show how dysbiosis negatively impacts intestinal motility, favouring constipation and the development of other intestinal pathologies such as irritable bowel syndrome [16,57].

The permeability of the intestinal barrier is also regulated by the microbiota. Specifically, factors such as a deficit of tight-junction proteins (such as ZO-1) and a reduction in the thickness of the intestinal mucus layer are linked to increased permeability and inflammation [58,59]. This allows the translocation of molecular patterns associated with pathogens (PAMPs) and bacteria, which can trigger local inflammation and contribute to metabolic disorders [16,57,60].

## 2. Gut–Brain Axis

The gut–brain axis is a bidirectional system that connects the gastrointestinal (GI) tract with the central nervous system (CNS). This system integrates not only nervous signals, but also immunological, endocrine and metabolic signals. Its main function is to maintain body homeostasis, regulating processes such as digestion, emotional state, immunity and neuroplasticity. This axis allows the brain to act on the gut and the gut to respond by sending signals to the brain. It is composed of: (A) Autonomic nervous system (ANS), comprising the sympathetic (SNS) and parasympathetic (PNS) systems, with the vagus nerve playing a key role in the latter. The ANS regulates functions such as intestinal blood flow, gastrointestinal tract motility and gastric secretion, among others. (B) Enteric nervous system (ENS), often dubbed ‘the second brain’, which is composed of more than 100 million neurons distributed along the gastrointestinal tract and, although it is closely connected to the vagus nerve and thus to the CNS, it is able to operate autonomously. (C) Neuroendocrine system, consisting of the hypothalamic–pituitary–adrenal (HHA) axis, which coordinates stress responses and regulates the release of glucocorticoids that influence the gut–brain axis. (D) The intestinal immune system, composed of Peyer’s plaques, dendritic cells and lymphocytes, which in turn act as sensors for changes in the microbiota or infections [61,62,63]. Peyer’s plaques are accumulations of lymphatic tissue present mainly in the ileum of the small intestine. Their main function is to act as defensive barriers, modulating interactions with the gut microbiota. Peyer’s plaques are connected to the CNS via the enteric nervous system (ENS) and are responsible for processing microbial antigens and promoting the activation of regulatory T-lymphocytes (T-reg) and anti-inflammatory cytokines such as IL 4 and IL-10. This response can be communicated to the brain via the bloodstream or vagus nerve. Conversely, in situations of inflammation or dysbiosis, Peyer’s plaques may contribute to increased intestinal permeability, facilitating the passage of toxins and bacteria, as well as pro-inflammatory cytokines (IL-6, TNF-α) into the systemic circulation, having deleterious effects at the brain level [61,62,63,64]. Herein lies the complexity of this axis, as it is an interconnected network with multiple levels of regulation that depend on the balance between the microbiota, the immune system, the vagus nerve and the brain.

Although the gut–brain axis has been known for a long time, about two decades ago, due to the rise of microbiota research, the term ‘microbiota–gut–brain axis’ (MGBA) came to be used, and the microbiota was considered an essential element of this axis (Figure 1).

Having presented the components of the MGBA, it is necessary to understand how they are interrelated. The main direct communication pathway between the gut and the brain is the vagus nerve, as mentioned above. Enteroendocrine cells in the gut respond to mechanical stimuli (such as distension) or chemical stimuli (such as microbial metabolites) and activate vagal afferent pathways via SCFAs, the neurotransmitters gamma-aminobutyric acid (GABA) and serotonin (5-HT), and gut hormones. The information then travels to the nucleus of the solitary tract in the brainstem. From this nucleus, gut signals will eventually be distributed to the cerebellar amygdala, insula and prefrontal cortex primarily, thus affecting emotional state, appetite regulation, gastric secretion and gastrointestinal motility. In summary, via efferent pathways, the vagus nerve modulates digestive activity, regulating the muscle tone of the intestinal walls, the secretion of digestive enzymes and local circulation. Several studies have shown that stimulation of the vagus nerve can reduce activation of the HHA axis, improving stress responses and reducing symptoms of anxiety and depression. Other studies in germ-free mice also show that the absence of microbiota can alter the expression of genes related to neuronal plasticity in regions such as the hippocampus [61,62,63].

The blood–brain barrier (BBB) is a critical structure in regulating the passage of molecules from the bloodstream to the CNS, ensuring a stable and protected environment for the brain. Its modulation occurs through complex interactions between endothelial cells, astrocytes, pericytes, microglia and the gut microbiota, which play a key role in the development and maintenance of the BBB, especially early in life. The BBB is not a static structure, but a highly dynamic system that responds to local (CNS) and systemic (microbiota and immune system) signals. When intestinal dysbiosis occurs, harmful metabolites such as lipopolysaccharide (LPS) and PAMPs are generated, which, when crossing the BBB and activating microglia, induce a neuroinflammatory response with subsequent increased permeability. This allows access of monocytes and lymphocytes, which accentuates inflammation of the nervous tissue and consequently the resulting damage [61,62,63]. In contrast, SCFAs and the ‘anti-inflammatory cholinergic pathway’ of the vagus nerve, SCFAs strengthen the cohesion between endothelial cells via claudin-5 and occludin, preventing the passage of immune cells and thus reducing systemic inflammation. Moreover, astrocytes and pericytes release trophic molecules such as vascular endothelial growth factor (VEGF) or transforming growth factor beta (TGF-β) that positively regulate BBB permeability [61,62,63].

Finally, there is an endocrine communication pathway. Enteroendocrine cells that release peptides such as ghrelin, leptin and peptide YY, which modulate appetite and reward responses in the brain [61,62,63].

Ultimately, two major information pathways can be highlighted: one upstream to the brain, consisting of the microbiota, vagus nerve, immune mediators and microbial metabolites; and one downstream to the gut, consisting of the ANS and HHA. The MGBA operates under a circular model: the brain can alter the gut environment through stress or autonomic dysfunction, which in turn affects the composition and function of the microbiota. This change generates metabolites and immune signals that feed back to the brain, perpetuating a cycle of dysfunction if the balance is not maintained. Thus, the bidirectional nature of this axis allows for rapid adaptations to internal and external changes but also makes it vulnerable to systemic dysfunction [61,62,63].

## 3. Gut–Brain–Microbiota Axis Approach to Neurological Diseases

Gut dysbiosis can trigger or aggravate neurodegenerative processes through multiple interconnected mechanisms. First, the gut microbiota produces a wide range of metabolites, including SCFAs, neurotransmitters and other neuroactive compounds derived from the metabolism of amino acids such as tryptophan. These metabolites have the capacity to modulate neuronal function and synaptic plasticity, influencing the synthesis and release of neurotransmitters such as serotonin, dopamine and gamma-aminobutyric acid (GABA) [61,65].

Another relevant aspect is the role of dysbiosis in increasing intestinal permeability and, consequently, in altering the BBB. A leaky gut allows the passage of endotoxins, which induces the production of proinflammatory cytokines and promotes neuroinflammation. This chronic inflammatory state in the CNS is a crucial factor in the pathogenesis of several neurological diseases, including anxiety, Alzheimer’s, Parkinson’s, multiple sclerosis (MS) and amyotrophic lateral sclerosis (ALS) [65].

For example, *Campylobacter jejuni* infection appears to increase anxious behaviour through activation of c-Fos proteins [16]. Conversely, experimental studies have shown that administration of certain probiotics such as *Lactobacillus* and *Bifidobacterium*, modify the expression of GABA receptors in brain areas linked to emotional regulation, suggesting a direct relationship between microbial composition and mood-related behaviours such as anxiety [16,65].

Alzheimer’s disease is characterised by the accumulation of senile plaques composed of beta-amyloid (Aβ) protein deposits in the brain and the formation of neurofibrillary tangles of hyperphosphorylated tau protein [16]. Several studies have confirmed the importance of the gut microbiota in the pathogenesis of this disease, with a decrease in *Firmicutes* and *Bifidobacterium* phyla compared to healthy subjects. Furthermore, through pro-inflammatory mechanisms, they may favour the production and accumulation of Aβ, altering synaptic function and exacerbating neurodegeneration [16,65].

Parkinson’s disease is a neurodegenerative disorder characterised mainly by motor symptoms such as bradykinesia, rigidity and resting tremor. These result from the loss of dopaminergic neurons in the substantia nigra of the brain and the accumulation of Lewy bodies, which contain α-synuclein [65,66]. During the pre-symptomatic stages of this disease, α-synuclein-mediated Lewy body pathology has been observed in the SNE and in the dorsal motor nucleus of the vagus nerve. The presence of Lewy bodies and abnormal accumulations of α-synuclein protein in these areas suggests that Parkinson’s may begin by affecting the gut and PNS before involving the brain, which may explain some of the early non-motor symptoms of Parkinson’s, such as digestive problems [16,65]. As in Alzheimer’s disease, alterations in the composition of the microbiota may trigger an inflammatory environment that accelerates the apposition of α-synuclein, thus contributing to the progression of this disease.

Multiple sclerosis (MS) is an autoimmune disease that causes demyelination and alterations in synapses. In fact, both MS patients and murine models of experimental autoimmune encephalomyelitis (EAE) have gut dysbiosis, which contributes to a decrease in the abundance of anti-inflammatory bacteria and an increase in species that promote inflammation, which may contribute to disease progression and exacerbation of symptoms [65,67].

Amyotrophic lateral sclerosis (ALS) is characterised by progressive degeneration of motor neurons. The relationship between gut microbiota and ALS is under investigation, as altered microbiota may favour the production of neurotoxins and the activation of systemic inflammatory responses that aggravate neuronal degeneration [65].

## 4. Gut–Brain–Microbiota Axis and Its Relationships with Stroke

Stroke is the second leading cause of death worldwide and the third leading cause of death and disability combined (measured in disability-adjusted life years, DALYs) [68,69,70]. Each year, more than 12.2 million new strokes occur, affecting 1 in 4 people in their lifetime [68,71].

The incidence of stroke is directly proportional to population ageing. This phenomenon is particularly relevant due to the global increase in life expectancy, especially in middle- and high-income countries. According to United Nations data, by 2050 the proportion of people over 65 years of age will double worldwide [69,70] and this could translate into a parallel increase in the incidence of stroke [68,69]. On the other hand, the average age of stroke onset has advanced, with an estimated 62% of cases occurring in those under 70 years of age [68].

Stroke is defined as a neurological deficit attributed to an acute focal CNS lesion of vascular cause. Ischaemic strokes account for approximately 80–87% of cases and are due to reduced blood flow, usually as a result of arterial occlusion. Venous infarction due to occlusion of cerebral veins or venous sinuses may also occur, although much less frequently. Ischaemic stroke is distinguished from transient ischaemic attack by the presence of an infarct on brain imaging. Traditionally, patients whose symptoms are resolved in less than 24 h have been referred to as ischaemic stroke. However, we now know that such patients show evidence of infarction on diffusion-weighted magnetic resonance imaging (MRI) in approximately 40% of cases and represent a high-risk group for recurrent stroke [71]. Occlusion of the blood vessel generates cerebral ischaemia, which progresses to infarction if flow is not restored. Surrounding the infarct core is the ‘ischaemic penumbra’, a region with depressed neuronal activity but potentially recoverable by reperfusion treatments such as intravenous thrombolysis or thrombectomy [71,72]. The main causes of the obstruction are:Embolism is the most common mechanism of stroke. The vast majority of emboli are generated from the heart due to cardiac disease, such as atrial fibrillation, valvular heart disease and acute myocardial infarction [70,72].Large vessel disease occurs mainly in the proximal cervical internal carotid arteries, with arterial dissection due to trauma being the most common cause in young people [70,72].Small vessel disease mainly affects the penetrating arteries and often causes small deep strokes, known as lacunar infarcts. These arteries are more vulnerable to the effects of chronic hypertension, unlike the large vessels [72].

The remaining 10–20% of strokes are haemorrhagic and result from rupture of cerebral arteries, leading to intracerebral or subarachnoid bleeding. The main causes include uncontrolled hypertension, cerebral amyloid angiopathy and vascular malformations such as aneurysm [71,73].

Estimates indicate that approximately 90% of strokes are attributable to modifiable risk factors [71,74]. The remaining 10% are due to non-modifiable factors, including older age (which remains the most important risk factor for stroke) [75], black race [70], male gender (although because women live longer, their lifetime stroke rate is higher) [75] and family history [70].

The main modifiable risk factors for stroke are: [70].

Arterial hypertension (AHT). It is considered the most relevant risk factor for stroke, both for ischaemic stroke and intracerebral haemorrhage [70,71,75].Diabetes mellitus type 1 (T1DM) and type 2 (T2DM). They are associated with twice the risk of stroke and increase the mortality rate by approximately 20% [70,75].Obesity. The risk of stroke is directly proportional to the increase in body mass index (BMI) and is partly due to indirect complications such as hypertension and dyslipidaemia [75]. This association is of great relevance as it brings forward the age of onset of cardiovascular disease [76].Atrial fibrillation (AF). AF increases the risk of stroke by up to five times and contributes to 15% of all strokes [70,71].Hyperlipidaemia. Recent studies associate low HDL levels (<0.90 mmol/L) as well as high total triglyceride levels (>2.30 mmol/L) with an increased risk of stroke, as well as a doubling of the risk of death [70,76].Oral contraceptives and hormone replacement therapy also increase the risk of stroke through thromboembolic mechanisms. A recent meta-analysis has suggested that each 10 µg dose increase and each additional 5 years of use increases the risk of stroke by 20% [76].Alcohol and drug abuse. The risk of ischaemic stroke is related to the amount of alcohol consumed daily. However, any alcohol consumption increases the risk of haemorrhagic stroke. On the other hand, regular use of phenethylamines (such as cocaine) is associated with an increased risk of all stroke subtypes and is a predisposing factor in those under 35 years of age [76].Smoking: Smoking is directly related to an increased risk of stroke and contributes to 15% of stroke mortality, especially in young patients [75,76].

Recently, several studies have also shown that an altered gut microbiota may also be a risk factor for stroke [74,77]. In addition, gut bacteria play a key role in the formation of atherosclerotic plaques and produce neuroactive compounds that may modulate neuronal function. Several reports have revealed the impact of ischaemic stroke on gut dysfunction and dysbiosis, highlighting the complex relationships between brain, gut and microbiota [74].

### 4.1. Microbiota and Atherosclerosis

The interaction between gut microbiota and atherosclerosis has emerged as an area of growing interest, as they can decisively influence the formation of atheromatous plaques and, ultimately, the development of thrombotic events such as ischaemic stroke.

The Wuhan group (China) studied the relationship between different bacterial species and atherosclerosis in various locations (cerebral, coronary and peripheral). The results of this study showed that certain bacteria, such as the genus *Ruminiclostridium*, have a protective effect on cerebral atherosclerosis, while others, for example, belonging to the *Rikenellaceae* and *Streptococcaceae* families or genera such as *Paraprevotella* and *Streptococcus*, are pathogenically associated with increased risk. In the field of coronary and peripheral atherosclerosis, protective microorganisms (such as some members of the *Acidaminococcaceae* family and the genus *Desulfovibrio*) and others that play a deleterious role were identified, showing that the influence of the microbiota is specific to a given vascular location. These findings indicate that the composition and diversity of the gut microbiota may directly modulate plaque formation in arteries and, consequently, determine the risk of serious cardiovascular events [78].

A review published by a group from Xiamen (China) explores the mechanisms by which microbiota-derived metabolites affect the stability of atheromatous plaques. These metabolites include trimethylamine-N-oxide (TMAO), lipopolysaccharide (LPS), phenylacetylglnutamine (PAGln) and SCFAs [79,80]. For example, TMAO has been implicated in platelet activation and induction of inflammatory responses, resulting in damage to the vascular endothelium and in weakening the structure of the fibrous cap lining the plaque, triggering thrombus formation [80].

On the other hand, LPS, a component of the cell wall of Gram-negative bacteria, is released in the gut after the death of the bacteria. When the integrity of the intestinal barrier is compromised, LPS can pass into the systemic circulation. Once in the bloodstream, it activates Toll-like receptors (TLR), specifically TLR2 and TLR4 receptors, triggering a chronic inflammatory response that contributes to the instability of atheromatous plaques [79,80].

In addition, metabolites such as PAGln and SCFAs also play important roles. While PAGln has been linked to prothrombotic effects, SCFAs have shown stabilising properties, maintaining the integrity of the fibrous cap and thus conferring a lower likelihood of plaque rupture [80] (Figure 2).

### 4.2. Microbiota and Immunosuppression

After stroke, intestinal microbial dysbiosis occurs, characterised by an increase in proinflammatory bacteria, such as Enterobacteriaceae and Firmicutes, and a decrease in SCFA-producing species, such as *Prevotella* and *Ruminococcus* [64,79,81]. This alteration of bacterial diversity favours the overgrowth of opportunistic pathogenic microorganisms, affecting the post-ictal homeostasis of the organism. Intestinal dysbiosis leads to increased intestinal permeability, which favours the growth and translocation of opportunistic bacteria such as *Streptococcus pneumoniae*, *Klebsiella pneumoniae*, *Pseudomonas aeruginosa*, *Escherichia coli* and *Enterococcus faecalis* across the intestinal barrier, reaching sterile organs such as lungs, spleen and brain, thus favouring the development of stroke-associated infections (SAIs) [64,81].

At the metabolic level, bacterial dysbiosis results in a decrease in SCFAs and an increase in LPS, favouring the activation of TLR4 receptors in astrocytes and the production of proinflammatory cytokines that would end up destabilising the BBB and forming cerebral oedema. In parallel, brain damage triggers the release of PAMPs, which promotes the migration of T-helper (Th) lymphocytes (Th1/Th17) who are proinflammatory, towards the ischaemic brain tissue, intensifying the inflammatory response and enlarging the injured area [64].

Following the acute inflammatory response, the HHA axis and the sympathetic nervous system (SNS) are activated, leading to a state of systemic immunosuppression. Activation of the adrenal axis increases the release of glucocorticoids, which inhibit T-lymphocyte activity, suppress the production of pro-inflammatory cytokines (IL-1β, IL-6, TNF-α) and reduce the function of antigen-presenting cells. In turn, hyperactivation of the SNS increases the production of catecholamines, which induce lymphocyte apoptosis (CD4+ T cells).

## 5. Microbiota and Stroke: Microbiota Profiles and Their Relationship to Stroke

### 5.1. Ischaemic Stroke

Research on the relationship between gut microbiota and stroke is still in its infancy, but several experimental studies have identified specific microbiota profiles that are associated with an increased risk of stroke. In addition, some clinical research also supports this process, as approximately 50% of stroke patients have gastrointestinal symptoms [81].

Analysis of faecal samples has revealed changes in the composition of the gut microbiota, with alterations in the proportions of *Firmicutes* and *Bacteroidetes* phyla, with an increase in *Firmicutes* being an independent predictor of ischaemic stroke risk [81]. Patients with ischaemic stroke were also found to have a higher abundance of certain proinflammatory bacteria, such as *Streptococcus*, *Lactobacillus*, *Prevotella*, *Paraprevotella* or in more severe cases *Proteobacteria* [81,82,83], while they had lower numbers of SCFA-producing bacteria, such as *Butyricicoccaceae*, *Barnesiella*, *Clostridiaceae* and *Lachnospiraceae*, which appear to have a protective effect against stroke [81,82,83,84].

However, a study focused on determining microbiota profiles related to large vessel disease identified five specific groups, including the genera *Bifidobacterium*, *Butyricimonas*, *Blautia*, and *Dorea* [85]. The abundance of these was significantly associated with stroke severity (according to the National Institutes of Health Stroke Scale (NIHSS)) and also with lipid and glucose metabolism. High triglyceride (TG) levels may be associated with a decrease in beneficial bacteria such as *Bifidobacterium* and *Blautia*, which favours a pro-inflammatory environment in the gut and brain. Thus, an increase in these in the context of hyperlipidaemia could be an attempt by the body to compensate for inflammation, but if TG levels are too high, the inflammatory response may continue to predominate, worsening neurological prognosis and reflected in higher NIHSS scores [85,86]. Other studies focusing on stroke severity found that *Streptococcus* [86] as well as *Collinsella*, *Ruminococcaceae*, *Akkermansia*, *Eubacterium oxidoreducens* and *Verrucomicrobiaceae*, are associated with a worse functional prognosis [87]; while *Escherichia-Shigella*, *Bacteroides* and *Agathobacter* [86], as well as *Lactococcus*, *Ruminococcaceae*, *Peptostreptococcaceae* and *Odoribacter* correlated with better clinical outcomes after ischaemic stroke [87].

### 5.2. Haemorrhagic Stroke

In this type of stroke, research is even more limited. A study by Wang et al. detected an increase in *Enterococcoccaceae*, *Clostridiales* and *Peptoniphilaceae* bacteria, along with a decrease in *Bacteroidaceae*, *Ruminocococcaceae*, *Lachnospiraceae* and *Veillonellaceae*. A negative relationship was also observed between Enterococcus abundance and neurological prognosis, while *Lachnospiraceae* status was associated as an independent predictor of neurological outcome [81]. Another study highlighted the *Porphyromonadaceae* family for their enormous negative impact, as they can damage the vascular endothelium, increasing the risk of haemorrhage. The possible involvement of *Pasteurellales* was also highlighted, although further studies are needed [83].

### 5.3. Effects of Stroke on the Gut Microbiota

Ischaemic stroke induces intestinal dysbiosis characterised by a loss of microbial diversity, a decrease in neuroprotective metabolites (such as SCFAs) and an increase in pro-inflammatory bacteria. These alterations contribute to systemic immune activation and secondary neuroinflammation, affecting the integrity of the BBB and hindering functional recovery [88]. It has been proposed that post-stroke cognitive impairment (PSCI) is related to changes in the microbiota. In patients with PSCI, significant increases in pro-inflammatory bacteria such as *Enterobacteriaceae*, *Streptococcaceae*, and *Lactobacillaceae* are observed, as well as decreases in short-chain fatty acid (SCFA)-producing microorganisms such as *Oscillibacter*, *Ruminococcus*, and *Coprococcus*. SCFAs are crucial for cognitive function and synaptic plasticity, and their deficiency may contribute to neuroinflammation and cognitive impairment [89]. Table 1 compiles the main bacterial taxa of the gut microbiota and their effects on stroke.

## 6. Links Between Microbiota and Blood Biomarkers Related to Increased Stroke Risk

Ischaemic stroke is a multifactorial entity involving inflammatory, metabolic, neurovascular and immune processes. In this context, blood biomarkers are emerging as useful tools both for risk stratification and for the prediction of post-stroke functional and cognitive evolution. In parallel, the gut microbiota has emerged as a key element in modulating many of the pathophysiological pathways involved. Due to the influence of the gut–brain axis in the pathogenesis of stroke, and especially the key role of the gut microbiota and its metabolites, it has been reported that certain molecules may act as biomarkers of stroke risk. Understanding the interactions between gut microbiota, blood metabolites and stroke risk could contribute to better prediction and management of stroke, reducing its impact on overall health [83].

### 6.1. Glutamate

Glutamate is the most abundant non-essential amino acid and the main excitatory neurotransmitter in the nervous system. It is a precursor of GABA, the main inhibitory neurotransmitter, which is essential for balancing excitatory signals and maintaining brain stability [90].

Glutamate may act as a neurotransmitter also in the gut, as expression of glutamate transporters (EAATs) has been found in enteric neurons. Moreover, glutamate is also important at the bacterial level as D-glutamate is an essential component of the bacterial peptidoglycan cell wall and is produced by lactic acid bacteria such as *Lactobacillus plantarum*, *Lactococcus lactis* and *Lactobacillus paracasei*. In addition, the enzyme Glutamate-decarboxylase (which allows transformation to GABA) is present in *Corynebacterium glutamicum*, *Brevibacterium lactofermentum*, *Mycobacterium smegmatis* and *Bacillus subtilis*. This explains why alterations in the gut microbiota may affect the Glu-Gln-GABA (glutamate-glutamate-GABA) cycle, which could contribute to the development of depression and other neuropsychiatric disorders [90].

A key marker of stroke is a dramatic increase in glutamate in the extracellular fluid (ECF) and cerebrospinal fluid (CSF). This is because during stroke, glutamate transporters EAATs (which normally remove excess glutamate) can reverse their function and release glutamate into the extracellular space, contributing to neuronal death by excitotoxicity. These processes increase activation of NMDA and AMPA-kainate receptors, exacerbating neuronal injury. In addition, a direct correlation has been found between blood glutamate levels and brain oedema, suggesting a key role for glutamate in the progression of brain damage [90].

Up to one third of stroke survivors develop post-stroke depression (PSD) within the first five years. This is associated with the size and location of the brain lesion, particularly in the left frontal cortex and basal ganglia, which regulate mood [90].

### 6.2. Trimethylamine N-Oxide (TMAO)

TMAO production has been linked to several adverse health effects. In particular, elevated levels of TMAO are strongly associated with an increased risk of atherosclerosis, myocardial infarction and stroke [91,92,93,94], as it participates in the deposition of cholesterol on arterial walls, enhances platelet aggregation, suppresses reverse cholesterol transport and causes vascular dysfunction, thus resulting in a very pro-atherogenic effect [94].

TMAO is a metabolite produced from the diet, mainly from red meat, seafood, eggs and dairy products containing nutrients such as choline, betaine and carnitine [91,92,93,94]. Gut microbiota, in particular species belonging to the phyla *Firmicutes*, *Proteobacteria*, *Lachnoclostridium*, *Clostridium* and *Escherichia* and specifically six genera (*A. hydrogenalis*, *C. asparagiforme*, *C. hathewayi*, *C. sporogenes*, *E. fergusonii*, *P. penneri*, *P. rettgeri* and *E. tarda*), metabolise these substances into trimethylamine (TMA) (37,38) which is rapidly absorbed into the blood and transformed into TMAO in the liver via the flavin monooxygenase (FMO) system 1 and 3 [91,92,93,94].

An increase in TMAO concentration may be due to diet, changes in the composition of the gut microbiota, or damage to the intestinal barrier [91,92,94]. For example, vegan and vegetarian diets are associated with lower plasma TMAO levels [94].

Recent studies have found that TMAO may be involved in the pathogenesis of diseases such as hypertension, diabetes, obesity-associated dyslipidaemia, atrial fibrillation, atherosclerosis (and thus thrombosis), ultimately increasing the risk of ischaemic stroke [91,92,93]. In a case–control study, it was shown that every 1 μM increase in TMAO increased the risk of severe ischaemic stroke by 22% [91].

Not only does TMAO increase the risk of stroke, but elevated levels of TMAO are associated with higher NIHSS scores, larger stroke size on MRI, worse prognosis and increased risk of recurrence [91,92,93]. It is also known that several microbial genes, such as the *CutC* gene, or the *CntA* gene may influence the production of TMA and TMAO, which could impact stroke severity and lead to unfavourable functional outcomes [92,94].

In addition, TMAO can cross the BBB and worsen post-stroke inflammation by activating the NLRP3 inflammasome, which promotes the release of the proinflammatory cytokines IL-1β and IL-18, reactive oxygen species (ROS), as well as the activation of microglia and astrocytes [91,92,93]. This process affects endothelial permeability and contributes to BBB disruption, which may be related to haemorrhagic stroke (HFI) [93].

TMAO may also be implicated in small vessel disease of the brain, as elevated levels are associated with increased white matter damage [91,92,93]. In a study conducted by Shandong University (China), a positive correlation was found between plasma TMAO levels and the severity of leukoencephalopathy, as well as a strong predictive ability [92].

In addition to conventional treatments, it is suggested that therapy targeting the TMAO pathway could be a strategy to reduce the risk of recurrence and improve the prognosis of stroke [91]. Although precise quantification remains a challenge, evidence suggests that the risks associated with TMAO are significantly lower than those posed by saturated fats in terms of contributing to atherosclerosis [94].

### 6.3. Phenylacetylglnutamine (PAGIn)

Phenylacetylglnutamine is a highly nitrogenous metabolite produced by the conjugation of feline acetic acid (PAA) with glutamine in the liver and kidneys [95,96].

*Christensenellaceae*, *Ruminococcaceae*, *Lachnospiraceae*, *Proteus mirabilis*, *Acinetobacter baumannii*, *Klebsiella pneumoniae*, *Bacteroidetes*, *Firmicutes*, *Proteobacteria* and *Staphylococcus aureus* are some of the main microorganisms involved in the production of PAA from phenylalanine, an amino acid found naturally in eggs, milk and meat. Since PAGln is present in organs such as the kidney and liver and in the systemic circulation, elevated levels of this metabolite have been linked to the onset and progression of diseases affecting the renal, hepatic, cardiovascular and cerebrovascular systems [96].

PAGln has been shown to participate in thrombus formation by enhancing platelet function via G protein-coupled receptors (GPCRs) as well as with α2A, α2B and β2-adrenergic receptors. Further research has shown that specific silencing by ADR-siRNA inhibits PAGln-induced prothrombotic phenotypes [95,96]. Therefore, PAGln may play a role in acute ischaemic stroke and its study as a potential biomarker is important as it correlates with stroke severity [96].

### 6.4. Short-Chain Fatty Acids (SCFA)

SCFA comprise organic fatty acids comprising mainly acetic acid (acetate), propionic acid (propionate) and butyric acid (butyrate) in molar ratios of approximately 40:20:20. These compounds are mostly generated from the fermentation of fibre and proteins by intestinal microbiota (mainly *Prevotella*, *Bifidobacterium*, *Parasutterella* and *Roseburia* [97]). While most of the butyrate serves as an energy source for enterocytes, the remaining SCFAs reach the liver, where they are used in gluconeogenesis and lipogenesis [98].

During acute ischaemic stroke, alterations in the composition of the gut microbiota are observed, with reciprocal influences between the two. This relationship has been extensively studied, and most studies agree that patients with ischaemic stroke have significantly lower levels of acetate, propionate and butyrate compared to the healthy group. Furthermore, SCFA levels showed a negative correlation with the prognosis of ischaemic stroke. This negative correlation also exists with each of the various risk factors for ischaemic stroke, including HTN, hyperlipidaemia, T2DM, obesity, atherosclerosis and atrial fibrillation, highlighting the involvement of these acids in stroke prevention [98]. In a study in rats with pre-eclampsia, significantly increased faecal concentrations of acetic acid, propionic acid and valerate were observed compared to the control group. Furthermore, these alterations occurred in the preclinical stage of pre-eclampsia, suggesting that assessment of these metabolites could be of added value for early diagnosis and risk stratification in both women with chronic hypertension during pregnancy and stroke [97].

Following ischaemic stroke, a state of systemic inflammation is generated that ultimately compromises both the BBB and the gut barrier [98]. SCFAs, being able to cross the BBB, are involved in several key processes: they repair the brain–intestinal barrier, mitigate oxidative stress, reduce the neuroinflammatory response and suppress both autophagy and neuronal apoptosis [98].

Each of these mechanisms highlights the therapeutic potential of modulating the levels and activity of SCFAs to improve prognosis in patients with ischaemic stroke by influencing critical processes such as barrier integrity, oxidative response, inflammation and neuronal survival [98].

### 6.5. Neurochemical Biomarkers

#### 6.5.1. Brain-Derived Neurotrophic Factor (BDNF)

BDNF has gained importance in recent decades due to its involvement in neuroplasticity, neuroprotection and neuronal regeneration processes, with a key role in memory, learning and emotional state [99,100,101]. It is part of the neurotrophin family and has a high affinity for the TrkB receptor. Its activation inhibits apoptotic processes and promotes neuronal survival, synaptogenesis and angiogenesis [99]. In the context of ischaemic stroke, reduced cerebral blood flow triggers a cascade of events that culminates in neuronal death. Indeed, a recent meta-analysis by Mojtabavi et al. confirmed that stroke significantly reduces BDNF levels, affecting neuroplasticity and favouring post-stroke depression [102]. In this scenario, increased BDNF levels could act as a neuroprotective factor (Figure 3). Also, various elements that raise BDNF levels (such as physical activity, memantine, donepezil or statins) could indirectly exert a beneficial effect [100].

The relationship between gut microbiota and BDNF is not fully defined. SCFAs have been documented to modulate the expression of BDNF-related genes in brain areas such as the hippocampus. In addition, the microbiota is responsible for the production of neurotransmitters (5-HT, GABA and dopamine) and modulation of the immune system, thereby regulating BDNF expression [101].

Beyond its biological action, recent studies have pointed to BDNF as a prognostic biomarker in stroke, linking high serum BDNF concentrations in the acute phase of stroke with a lower incidence of subsequent severe disability and mortality [99].

#### 6.5.2. Glial Cell-Derived Neurotrophic Factor (GDNF)

GDNF is another key molecule in the processes of neuroprotection and tissue repair after focal ischaemic stroke. Initially, its neuroprotective effects were thought to be limited exclusively to dopaminergic neurons. It is now known to promote the survival of multiple neuronal types in various pathological contexts, including cerebral ischaemia [103]. Under physiological conditions, GDNF is predominantly expressed in neurons. However, following cerebral ischaemia, a marked induction of GDNF is observed in the area of reactive astrocytosis in the peri-infarct zone, a phenomenon that appears to be related to the processes of endogenous neuroprotection and spontaneous repair of brain tissue [103,104] (Figure 3). This has been demonstrated in studies with murine models where specific gene inhibition of GDNF in astrocytes resulted in increased brain infarct volume, increased number of impaired neurons and decreased neuronal proliferation in the peri-infarct zone [104].

#### 6.5.3. Glial Fibrillary Acidic Protein (GFAP)

GFAP is a key structural protein of the astrocyte cytoskeleton and one of the main markers of reactive gliosis, a key cellular response to acute brain damage in ischaemic stroke. In acute phases, it is involved in ionic homeostasis and the reuptake of neurotransmitters such as glutamate, while in subacute and chronic phases it promotes glial scar formation [105,106,107]. Furthermore, GFAP is not expressed outside the CNS, making it a highly specific and dynamic biomarker, accessible peripherally and with diagnostic, prognostic and therapeutic implications [105] (Figure 3).

The study by Forró et al. demonstrates that GFAP levels are significantly elevated 24 h and 7 days after acute stroke, with a subsequent progressive decrease. This suggests that gliosis following cerebral ischaemia is a dynamic process that varies throughout the course of the disease. A positive correlation with severity (NIHSS scale) and short-term functional capacity (mRS scale) was also found, showing its early prognostic value [105]. A second study by Li et al. investigated the relationship between GFAP and the occurrence of intracranial haemorrhage after endovascular thrombectomy. GFAP levels increased significantly in the first 24 h after thrombectomy, resulting in a 51.8% risk of cerebral haemorrhage. These findings allow for anticipation of complications and adjustment of therapeutic decisions [106].

#### 6.5.4. Neurofilament Light Chain (NfL)

NfL is a structural protein that forms part of the axonal cytoskeleton, whose main function is to maintain the stability and calibre of the axon, ensuring effective transmission of nerve impulses. In the event of axonal damage, such as occurs during stroke, NfL is initially released into the CSF and subsequently into the bloodstream. Thus, elevated blood levels of NfL indicate the presence of an active neuroaxonal injury, making it an ideal marker of stroke-associated neuronal damage [107,108] (Figure 3).

Personalised prediction of functional prognosis after severe acute ischaemic stroke is essential to compare different therapeutic strategies. Recent studies have demonstrated a direct proportional relationship between functional outcome and the biomarkers NfL and GFAP. In this regard, a study developed by the University of Würzburg (Germany), showed a significant correlation between serum levels of both markers with functional outcome and risk of mortality at 3 months after stroke. Specifically, it is recommended to measure GFAP in the first 24 h and NfL between day 3 and 7, taking advantage of their different temporal dynamics [108].

#### 6.5.5. Neuregulin-1 (NRG-1)

Neuregulins (NRGs) are a family of glycoproteins belonging to epidermal growth factors (EGFs). These proteins are expressed in multiple isoforms (from NRG1 to NRG4) and are characterised by containing a specific domain for receptors of the ErbB family. The activation of these receptors triggers intracellular signalling cascades (such as the PI3K/AKT and MAPK pathways), responsible for regulating proliferation, differentiation, cell survival and synaptic plasticity. In the nervous system, NRG1 is essential for myelination of peripheral axons by Schwann cells and synaptogenesis. Its dysfunction has been mainly associated with neurodegenerative diseases (such as ALS and Alzheimer’s). In addition, it has recently gained importance in stroke thanks to its therapeutic potential, as it can cross the BBB and maintain the functionality of damaged nervous tissue. In preclinical models, NRG-1 has been shown to reduce neuronal death and inflammation after stroke by up to 90%, being effective up to 12 h after the ischaemic event, which significantly extends the therapeutic window [109].

The gut microbiota, by influencing the development of the SNP, can modulate the expression of NRG-1. In a study carried out in germ-free mice, an alteration in the development of peripheral nerves was observed, characterised by axons of smaller diameter. In addition, these animals showed an overexpression of the type III isoform of NRG-1, suggesting a possible compensatory mechanism against poor Schwann cell development and myelination [110].

#### 6.5.6. Nerve Growth Factor (NGF)

NGF is a neurotrophin essential in neuronal survival, differentiation, and regeneration. In ischaemic stroke, NGF has demonstrated remarkable therapeutic potential by modulating key processes such as apoptosis, inflammation, neurogenesis, and angiogenesis.

During the acute phase of stroke, activation of TrkA receptors occurs, triggering the PI3K/AKT and MAPK signalling pathways. These pathways promote neuronal survival and synaptic stability. All this would be in a regenerative environment thanks to the inhibition of pro-inflammatory cytokines such as TNF-α and IL-1β. In the recovery phase, NGF enhances neurogenesis and axonal regeneration, facilitating the formation of new neural networks and functional recovery. At the same time, it stimulates angiogenesis through the regulation of VEGF, which improves blood flow in ischaemic regions and promotes tissue repair [111].

#### 6.5.7. Insulin-like Growth Factor-1 (IGF-1)

IGF-1 is a key molecule in the regulation of neurogenesis, synaptic plasticity, and cell survival. In the context of ischaemic stroke, low levels of IGF-1 are associated with a higher risk of suffering a cerebrovascular event, higher scores on severity scales (NIHSS, mRS), worse outcome, and higher mortality [112]. In contrast, in animal models, the administration of IGF-1 reduces the size of the cerebral infarction and improves motor, sensory and cognitive functions after stroke [112].

#### 6.5.8. S100B

S100B is a calcium-fixing protein, expressed exclusively in the CNS, where it is produced by astrocytes. It participates in processes such as the regulation of calcium homeostasis, neurogenesis or the regulation of the brain’s immune response. At low concentrations, S100B can have trophic effects, promoting cell survival. However, at high concentrations, it acquires a neurotoxic and pro-apoptotic role. In addition, after acute brain injury, the BBB is compromised, allowing S100B to pass into the bloodstream, where it can be measured. High levels of S100B in the blood in acute phases of stroke correlate with glial injury, greater extent of cerebral infarction, worse neurological outcome, and higher scores on scales such as NIHSS or mRS [107].

## 7. Conclusions and Perspectives

The microbiota plays an essential role in the body’s homeostasis, playing key roles in metabolism, immune regulation, the integrity of epithelial barriers and particularly in the gut–brain axis. Its dynamic and specific composition for each individual and in different situations makes it a determining factor in the pathogenesis of multiple diseases, including neurological, cardiovascular, metabolic and inflammatory diseases, among others.

Currently, the therapeutic options available for ischaemic stroke are limited to pharmacological thrombolysis and mechanical thrombectomy. Given that approximately 70% of strokes do not receive any type of specific treatment, there is a need for new approaches, both preventive and therapeutic [81]. In this context, the study of microbiota would also acquire a decisive role in the identification and validation of new targets on which to have an impact, given its potential usefulness for prophylaxis and/or treatment.

To modify the composition of the microbiota and make it less prone to the development of diseases (such as stroke), some strategies could be considered, such as the following.

### 7.1. Dietary Interventions

Diet is the most accessible factor among those that influence the quantity and diversity of the gut microbiota. High-fibre diet, such as Mediterranean diet or vegetarian diet, have been shown to promote intestinal homeostasis, reducing the *Firmicutes*/*Bacteroidetes* index and regulating the production of microbial metabolites (increasing the production of SCFAs and decreasing the production of TMAO) [64,81,113]. Conversely, diets high in salt, sugar, or saturated fats alter microbial diversity and increase inflammatory responses, exacerbating post-stroke brain damage [81,113].

SCFAs (acetate, propionate, and butyrate) are produced by the fermentation of fibre by the gut microbiota and have anti-inflammatory properties. A diet rich in fibre is able to reduce the risk of stroke by up to 10%. Because of this, other routes of administration are being attempted, such as direct SCFA supplementation with sodium butyrate, butyrate-rich stool transplantation, and encapsulated acetate [64]. On the other hand, TMAO is a microbial compound highly involved in the formation of atheromatous plaques. Both the vegan and Mediterranean diets can significantly reduce their levels, which would reduce the risk of suffering a stroke [64].

The MIND diet (‘Mediterranean-DASH Intervention for Neurodegenerative Delay’) is a combination of the Mediterranean Diet and the DASH diet (‘Dietary Approaches to Stop Hypertension’). It was first devised in 2015 from foods with neuroprotective properties, including: green leafy vegetables, antioxidant-rich fruits, legumes, fish and virgin olive oil as the main source of fat. At the same time, the consumption of red meat, fried meat, and ultra-processed foods is restricted [114,115,116].

A recent systematic review demonstrated the effectiveness of the three nutritional patterns (Mediterranean Diet, ketogenic and MIND) in preventing cognitive decline and/or improving mental abilities in patients with some degree of dementia. However, the Mediterranean Diet, being the most studied, stood out above the other two in terms of results [114].

On the other hand, a systematic review focused exclusively on MIND diet and cardiovascular diseases found a protective effect in this diet against cardiovascular risk factors, such as BMI, lipid profile or hypertension, among others. A preventive role was also observed in diseases such as T2DM, arteriosclerosis and stroke [115]. As for the latter, two studies conducted in Chicago (USA) supported these findings. The first showed that good adherence to the MIND diet reduced the risk of stroke by 59% [116], while the second showed that adherence to this diet reduced cognitive impairment in people with a history of stroke in the long term [117].

### 7.2. Probiotics-Prebiotics-Synbiotics

Probiotics (such as *Lactobacillus* and *Bifidobacterium*) are live microorganisms that promote the growth of beneficial bacteria, helping to maintain intestinal homeostasis [81,113]. Some of these bacteria have neuroprotective properties, which is why they are also known as psychobiotics. These compounds have been shown to improve post-stroke recovery and reduce hospital stay [64], thanks to functions such as the modulation of the HPA axis, GABA metabolism and the neurotrophic factor BDNF [113].

Prebiotics (such as indigestible fibres and bioactive compounds) are dietary components that also stimulate the growth of beneficial bacteria [81,113]. Among them, indigestible galactooligosaccharides and fructooligosaccharides promote the regulation of the HPA axis, neurotransmitters such as serotonin and LPS levels, improving the state of neuroinflammation [113].

As mentioned above, elevated levels of LPS promote a state of hypercoagulability that increases the risk of stroke. However, it has been seen that small doses of LPS prior to a stroke can have a neuroprotective effect by increasing brain tolerance to ischaemia. This opens the door to LPS as a possible preventive therapy against recurrences [64].

Synbiotics are combinations of probiotics and prebiotics that increase the amount of beneficial bacteria in the gut. They also perform the aforementioned functions, such as modulating the HPA axis and the immune system, or regulating LPS and BDNF levels [113].

These products are administered in the form of oral lyophilised products and can sometimes be given in combination (‘probiotic cocktails’), varying in quantity and composition depending on the objectives.

### 7.3. Faecal Microbiota Transplant

Faecal microbiota transplantation (FMT) is considered one of the most promising strategies for the treatment of gastrointestinal and neuropsychiatric diseases [81,113,118]. It consists of transferring complete or selected microbial components from a healthy donor to a recipient with the aim of restoring microbial balance (eubiosis) [113,118]. It can be administered by different routes, including rectal (endoscopy or enema) and oral (frozen faecal capsules) [113,118].

In the case of stroke, preclinical studies of FMT have been shown to reduce intestinal and brain inflammation, through the proliferation of *Lactobacillus*, the release of IL-10 and the increase in T-reg cells [81]. In animal models, FMT is capable of reversing intestinal dysbiosis after stroke, reducing systemic and brain inflammation (at the level of microglia), reducing infarct volume, and improving functional recovery [118,119]. In addition, FMT has been shown to significantly reduce the expression of pro-apoptotic and necroptotic markers and to increase the expression of Bcl-2 (an anti-apoptotic protein), preventing post-stroke neuronal death [119]. These findings suggest that FMT has considerable potential as a stroke treatment. However, more studies are required to validate the safety and effectiveness of this therapeutic resource in the long term [118,119].

### 7.4. Emerging Therapies

Various bioactive compounds of natural origin have been investigated, which, through the regulation of the intestinal microbiota, have neuroprotective effects. An example is the extract of *Periplaneta americana* (*PAS840*), which has been shown to improve ischaemic brain damage in rats by restoring the gut microbiota [120]. Another compound of interest is a bioactive peptide (KF6) derived from the fungus *Boletus griseus*-*Hypomyces chrysospermus*, capable of inhibiting the angiotensin-converting enzyme (ACE). In this way, it reduces blood pressure, a risk factor for stroke [121]. Finally, a study with an ultrasound-degraded polysaccharide of *Pueraria lobata* (PLP-3) shows that this compound can significantly attenuate brain damage following ischaemic stroke in mice through the inactivation of the proinflammatory pathway LPS-TLR4 in the brain [122].

In conclusion, current evidence suggests that the modulation of the gut microbiota and its metabolites (through dietary interventions, probiotics, faecal microbiota transplantation (FMT) or natural extracts) constitutes an emerging and promising therapeutic strategy in the approach to ischaemic stroke. Knowing the low toxicity of these interventions, they can be considered adjuvant strategies, complementing either preventive or therapeutic conventional treatments.

## Figures and Tables

**Figure 1 ijms-26-10071-f001:**
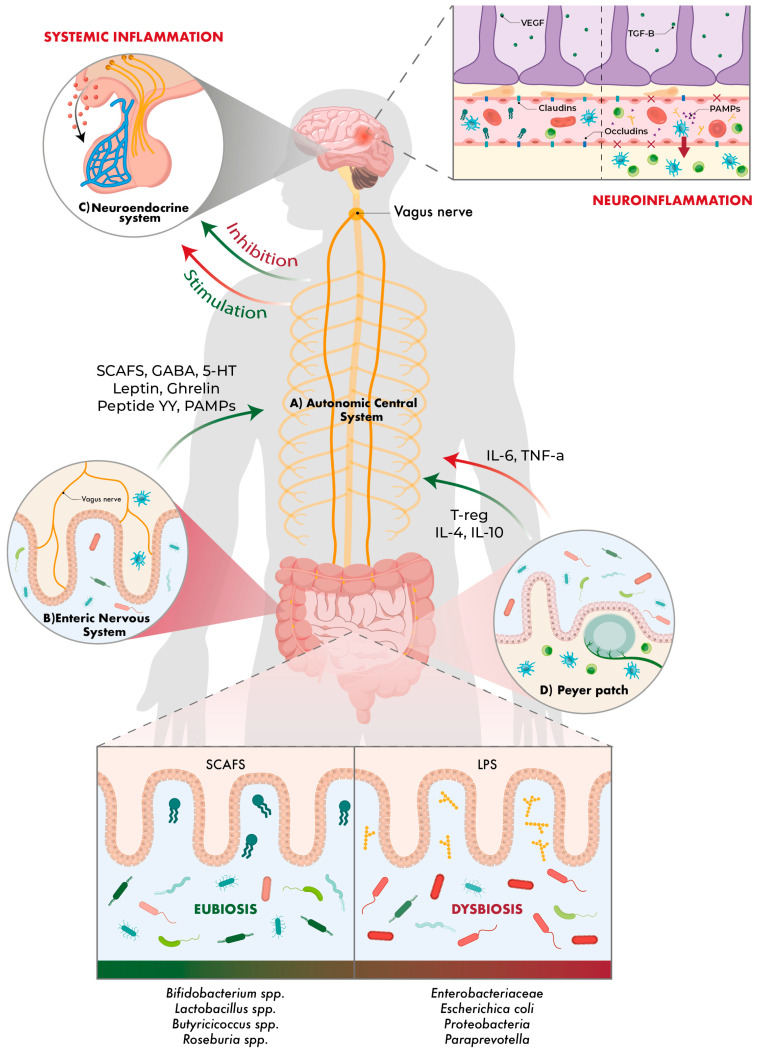
‘Gut–brain–microbiota axis’. The vagus nerve shows the bidirectional communication between the gut and the brain, as well as the bacterial taxa involved and the molecules that are modulated. Beneficial bacteria and effects are shown in green, and harmful ones in red.

**Figure 2 ijms-26-10071-f002:**
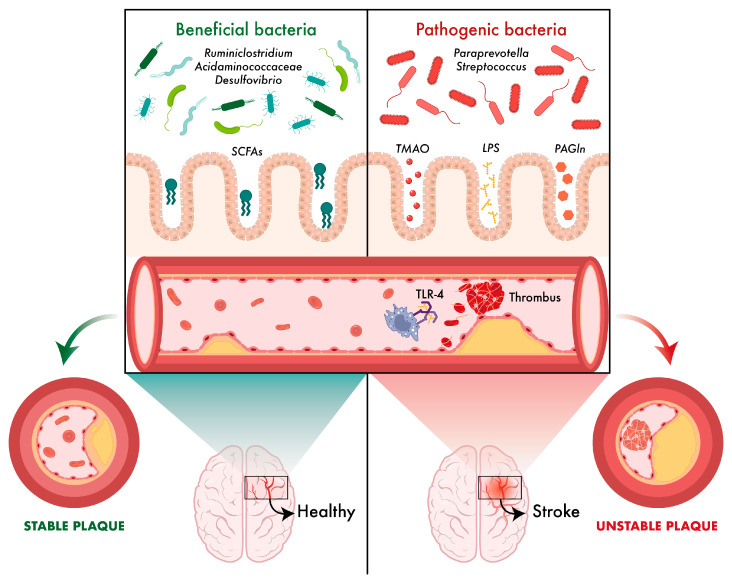
Role of microbiota in atherosclerosis. Inside the cerebral arteries, molecules and cells modulated by the gut microbiota are at work. Graphically is shown a vessel lumen in which beneficial bacteria and effects are shown in green and harmful ones in red.

**Figure 3 ijms-26-10071-f003:**
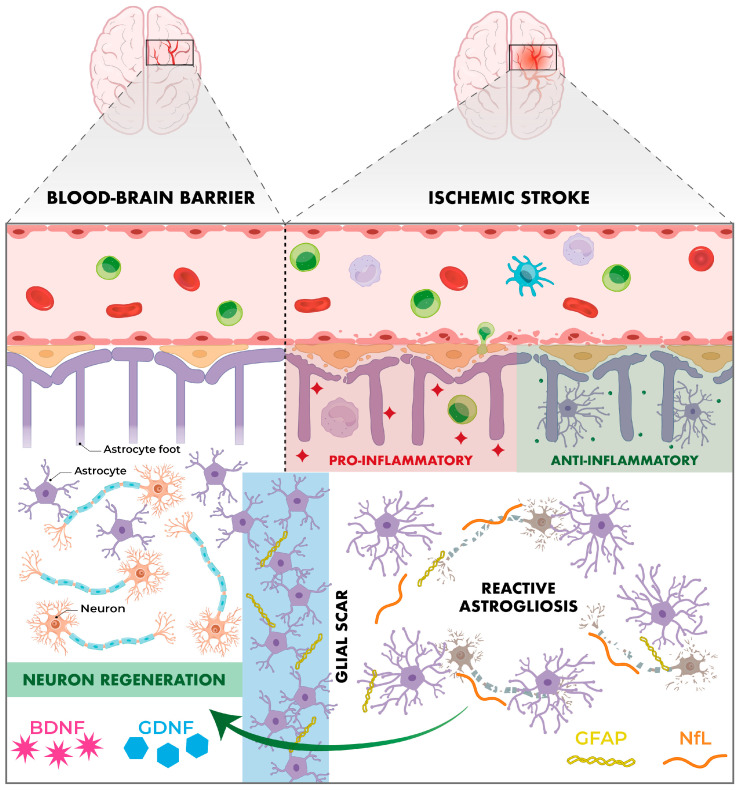
Post-stroke reactive astrocytosis. Some microorganisms in the gut microbiota can modulate (positively or negatively) the synthesis of neurotrophic factors and proteins in the axonal cytoskeleton and astrocytes, potentially inhibiting the neural regeneration process and promoting reactive astrogliosis.

**Table 1 ijms-26-10071-t001:** Summary of gut microbiota agents related to stroke and their actions.

Microorganism or Groups of Microorganisms	Overall Effect	Actions Related to Stroke
*Ruminiclostridium*	Beneficial	Protective effect against cerebral atherosclerosis; contributes to plaque stability and inflammatory reduction
*Rikenellaceae*, *Streptococcaceae*, *Paraprevotella*, *Streptococcus*	Detrimental	Associated with increased risk of atherosclerosis and thrombosis; promote vascular inflammation
*Prevotella*, *Ruminococcus*	Beneficial	Producers of SCFAs (butyrate); reduce inflammation and maintain the integrity of the intestinal and blood–brain barriers
*Enterobacteriaceae*, *Firmicutes*	Detrimental (if excessive)	Proinflammatory; increase intestinal permeability and immune activation via TLR4; worsen brain damage
*Streptococcus pneumoniae*, *Klebsiella pneumoniae*, *Pseudomonas aeruginosa*, *Escherichia coli*, *Enterococcus faecalis*	Detrimental	Opportunistic bacteria proliferate, leading to dysbiosis and an increase in post-stroke infections
*Butyricicoccaceae*, *Barnesiella*, *Clostridiaceae*, *Lachnospiraceae*	Beneficial	Producers of SCFAs (butyrate, propionate); modulate inflammation and protect the BBB
*Bifidobacterium*, *Blautia*, *Butyricimonas*, *Dorea*	Beneficial	Related to improved lipid metabolism and reduced inflammation; greater abundance correlates with improved prognosis
*Proteobacteria*	Detrimental	Increased in severe stroke; induces systemic inflammation and neuroinflammation
*Collinsella*, *Akkermansia*, *Eubacterium oxidoreducens*, *Verrucomicrobiaceae*	Detrimental	Associated with poorer functional recovery and greater neuroinflammation
*Escherichia-Shigella*, *Bacteroides*, *Agathobacter*, *Lactococcus*, *Ruminococcaceae* ^1^, *Peptostreptococcaceae*, *Odoribacter*	Beneficial	Related to better neurological prognosis and immune balance
*Desulfovibrio*, *Acidaminococcaceae*	Beneficial	Involved in lipid metabolism protective against atherosclerosis
*Porphyromonadaceae*	Detrimental	Endothelial damage and increased risk of cerebral haemorrhage
*Enterococcaceae*, *Clostridiales*, *Peptoniphilaceae*	Detrimental	Increase intestinal inflammation and worsen neurological prognosis
*Lachnospiraceae*, *Ruminococcaceae* ^1^	Beneficial	Predictors of better recovery; produce butyrate and reduce neuroinflammation

^1^ The same microorganism may be counted more than once if it appears grouped with other different microorganisms.

## Data Availability

No new data were created or analyzed in this study. Data sharing is not applicable to this article.

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
