# Peer review of "Molecular Mechanisms of the Microbiota–Gut–Brain Axis in the Onset and Progression of Stroke"

_ijms, 2025, doi:10.3390/ijms262010071_

Round 1

Reviewer 1 Report

Comments and Suggestions for Authors

In this article “Molecular mechanisms of the microbiota-gut-brain axis in the 2 onset and progression of stroke.” Authors have done a literature review about the bidirectional relationship between the brain and gut microbiota leading to the concept of the microbiota-gut-brain axis. Authors discuss how Intestinal dysbiosis (an imbalance in the gut microbiota) can promote a proinflammatory and prothrombotic state, as well as dyslipidaemia and dysglycemia, that increase atherogenic risk and consequently the risk of stroke. Dysbiosis can also lead to neuroinflammatory and neurodegenerative effects, compromising the integrity of the blood-brain-barrier and exacerbating brain injury after stroke. Specific bacterial profiles have been associated with varying levels of stroke risk, emphasising the role of gut microbiota-derived vasoactive metabolites such as TMAO, PAGln, and SCFAs, which may serve as biomarkers for stroke risk and severity. Gut microbiota also influences neurotrophic factors such as BDNF and GDNF (involved in recovery after stroke). Researchers explored the potential to modify the gut microbiota to either prevent stroke (by reducing risk) or improve outcomes (by decreasing severity and sequelae). Current scientific evidence supports the role of gut microbiota as a potential diagnostic and prognostic biomarker, as well as a therapeutic target. Authors have given extensive data about the subject. However, authors should include following in the revised manuscript for readers to understand and efficiently use the information in their research questions.

Please find my comments about the manuscript.

  1. Please use full form first in the text and then use short forms to understand well. For example, when using for the first time terms like SNE, PAMPs and so son.

  1. Authors MUST include a table mentioning all the microbiota that have been discussed in the review and add more details about microbiota which positively and negatively impact gut brain axis which will give a quick recap about everything in the review for the ease and increase of readership.

  1. Authors should also include more references in support of their review. Authors have been citing same references again and again. I am expecting more than 100 references for such review so that we are updating all the microbiota in this review.

  1. Authors in general have mentioned a very few studies about the involvement of microbiota in brain gut axis. Authors have explained about diseases or terms more rather than citing more literature about the evidence of microbiota in all the sections in review.

Please include all the changes in the revised manuscript.

Author Response

  1. Please use full form first in the text and then use short forms to understand well. For example, when using for the first time terms like SNE, PAMPs and so son.

Done. Except in the abstract (because if we did so, it would exceed the maximum number of words), in the rest of the sections we have specified what each acronym corresponds to each time it appears in writing for the first time, in those cases where it was not specified.

  1. Authors MUST include a table mentioning all the microbiota that have been discussed in the review and add more details about microbiota which positively and negatively impact gut brain axis which will give a quick recap about everything in the review for the ease and increase of readership.

Done. We have inserted an explanatory table (at the end of section 6) that can serve as a summary or brief guide. Thanks to this reviewer’s perspective, we believe that it is now easier for potential readers to understand, as the table can be used to outline the role of different microorganisms.

  1. Authors should also include more references in support of their review. Authors have been citing same references again and again. I am expecting more than 100 references for such review so that we are updating all the microbiota in this review.

Done. We have updated the bibliography and have now well exceeded 100 references, especially in the first part of the manuscript. We have not expanded the references in the section on biomarkers because there were fewer in the scientific literature and we had practically all of them in the previous version of the manuscript.

  1. Authors in general have mentioned a very few studies about the involvement of microbiota in brain gut axis. Authors have explained about diseases or terms more rather than citing more literature about the evidence of microbiota in all the sections in review.

Thank you for your comment. We have mentioned many more studies, especially regarding scientific evidence of the impact of microbiota on the brain and neurological diseases (and everything related to the microbiota-gut-brain axis). Thanks to this reviewer’s comment, the manuscript can gain strength by demonstrating that it is supported by numerous related scientific studies on this topic. Furthermore, if any potential readers wish to learn more, the new manuscript provides the possibility of locating these scientific studies. Thank you very much for this comment and all the previous ones. Thanks to them, we believe that the manuscript has improved in quality.

Reviewer 2 Report

Comments and Suggestions for Authors

The review is interesting and very extensive, but probably too long and needs to be significantly shortened because some paragraphs and even one chapter are out of context.

Major points

The sentence from lines 69 to 75 should be shortened.

Paragraph 4.1: This should be shortened to highlight a few points closely related to the objective of the review. Currently, it is merely a list of information that does not add any important insight.

Paragraph 4.2., from lines 368 to 376: This should be condensed as it contains some repetition.

Paragraph 4.2., from lines 384 to 390: The authors should better emphasize the significance of the changes they discuss.

Paragraph 5.1., from lines 406 to 420: This contains too much information, and most of the sentences are unclear. Please rewrite it to simplify and make it more readable.

Paragraph 6.4., from lines 541 to 543: The meaning of the sentence is unclear.

Paragraph 6.5. Neurochemical biomarkers.

All subsections, although interesting, are not related to the main objective and could be removed.

I remind the authors that GFAP is also expressed by enteric glia.

Section 7.1 should be shortened.

Minor points:

Row 43: likely is the 21st century

Row 50, remove ‘one’

Row 513: write ‘phenyl’ not feline

Row 529: delete ‘comprising’

Row 532: write ‘fibres’ not fibre

Author Response

Minor points to consider:

  1. Throughout section 6 (Links between microbiota and blood biomarkers…), authors should replace acronyms with their full words in level 2 headings. For example, replace “6.2 TMAO’ by “6.2 Trimethylamine N-oxide (TMAO)”.

Done. Except in the abstract (because if we did so, it would exceed the maximum number of words), in the rest of the sections we have specified what each acronym corresponds to each time it appears in writing for the first time, in those cases where it was not specified.

  1. Authors should include a legend for each figure.

Done. We have added a brief description of the figure as a caption. We believe not to make it too long because the concepts shown in each figure are also explained in detail in the text.

  1. Line 802: please, alphabetize a list of abbreviations.

Done. Now the abbreviations are displayed in alphabetical order. This makes them easier to find. Thank you very much for this comment.

  1. Please check DOI for references #27, #30, #35, and #52 and remove extra characters. That was probably done during formatting/editing the manuscript.

We have verified this both for the citations in the previous version of this manuscript and for the new references we have included in the current version. However, this is the format that appears when using the reference management computer programme (‘Mendeley'). One of the instructions given to us by the Editorial Assistants of the MDPI Group journals is not to manipulate references that are edited with a reference management application. Perhaps they have other computer tools with which they can modify the appearance and number of characters. If this defect is very important, we will report it directly to the Editorial Assistant staff.

Here are some specific comments:

  1. Introduction section
  2. a) There is only one level 2 heading in the introduction (“1.1. Role of the microbiota in the health-disease coupling”), is there a missing section in the introduction?

In fact, there is no missing section in the introduction, but rather we wanted the contents of section 1.1. (Role of the microbiota in the health-disease coupling) to appear at a different level, so that it is understood that it is a subsection of the introduction.

  1. b) The introduction focuses mainly on the gut microbiota, but it would be interesting to include information about stroke.

It maybe but regarding the genesis of strokes (atherogenesis) it could be considered that there is already text available, although we could also discuss general aspects of strokes, clinical characteristics, etc. Nevertheless the scope of this journal perhaps is mainly focused to molecular features of diseases. We have reviewed it again after reading your comment and we believe that the argument or main theme of the manuscript currently ranges from generalities about microbiota to its impact on stroke. If we were to develop a specific section on generalities about stroke, not only the common thread could be broken, but the text would also become very long and exceed the acceptable word count. However, in relation to your next observation, we have expanded the text of the manuscript, as we found your comment to be very accurate and interesting, and we have developed a new section in the manuscript that we think may be useful for potential readers (section 5.3. ‘Effects of stroke on the gut microbiota’). This section refers to molecular implications but also to post-stroke clinical manifestations when considering the effects of stroke on the gut microbiota.

  1. The microbiota gut-brain signaling is considered as bidirectional, is there any work investigating the consequences of a stroke on the gut microbiome and the gut function?

Thank you very much for this observation. This comment is very interesting because it helps us to improve the manuscript by adding an additional topic related to this field. It is well known that after a stroke there is a phase of immunosuppression. As a result, the lymphoid tissue that makes up Peyer's patches (which is the first line of immune response in the intestine) can be altered. We have found two articles focusing on this topic. Based on your observation, we have added the idea of the retrograde impact of stroke on the microbiota in section 5 (‘Microbiota and stroke: microbiota profiles and their relationship to stroke’) and we have added a new subsection at the end of this section, as mentioned above (section 5.3. ‘Effects of stroke on the gut microbiota’).

Reviewer 3 Report

Comments and Suggestions for Authors

Caballero-Villarraso et al. summarize in this review the current knowledge of the link between stroke and microbiota gut-brain axis. The article provides an interesting overview of the relationship between the pathogenesis of stroke and the microbiota composition and their derived metabolites. Finally, the authors highlight the emerging therapeutic strategies based on the modulation of the gut microbiota.

Minor points to consider:

  1. Throughout section 6 (Links between microbiota and blood biomarkers…), authors should replace acronyms with their full words in level 2 headings. For example, replace “6.2 TMAO’ by “6.2 Trimethylamine N-oxide (TMAO)”.
  2. Authors should include a legend for each figure.
  3. Line 802: please, alphabetize a list of abbreviations
  4. Please check DOI for references #27, #30, #35, and #52 and remove extra characters. That was probably done during formatting/editing the manuscript.

Here are some specific comments:

1.Introduction section        

a.There is only one level 2 heading in the introduction (“1.1. Role of the microbiota in the health-disease coupling”), is there a missing section in the introduction?      

b.The introduction focuses mainly on the gut microbiota, but it would be interesting to include information about stroke.

2.The microbiota gut-brain signaling is considered as bidirectional, is there any work investigating the consequences of a stroke on the gut microbiome and the gut function?

Author Response

(The authors gave the same response as above.)

Round 2

Reviewer 1 Report

Comments and Suggestions for Authors

Accepted as modifications have been done.

Reviewer 2 Report

Comments and Suggestions for Authors

The authors accepted many of the suggestions and explained why they couldn't accept any more.
The text has improved, although it's very long.